# Immunity to Varicella Zoster Virus in Healthcare Workers: A Systematic Review and Meta-Analysis (2024)

**DOI:** 10.3390/vaccines12091021

**Published:** 2024-09-06

**Authors:** Matteo Riccò, Pietro Ferraro, Salvatore Zaffina, Vincenzo Camisa, Federico Marchesi, Francesca Fortin Franzoso, Cosimo Ligori, Daniel Fiacchini, Nicola Magnavita, Silvio Tafuri

**Affiliations:** 1Servizio di Prevenzione e Sicurezza Negli Ambienti di Lavoro (SPSAL), AUSL-IRCCS di Reggio Emilia, Via Amendola n.2, 42122 Reggio Emilia, Italy; 2Occupational Medicine Unit, Direzione Sanità, Italian Railways’ Infrastructure Division, RFI SpA, 00161 Rome, Italy; pi.ferraro@rfi.it; 3Occupational Medicine Unit, Bambino Gesù Children’s Hospital IRCCS, 00165 Rome, Italy; salvatore.zaffina@opbg.net (S.Z.); vincenzo.camisa@opbg.net (V.C.); 4Department of Medicine and Surgery, University of Parma, Via Gramsci, 14, 43126 Parma, Italy; federico.marchesi@unipr.it (F.M.); francesca.fortin@unipr.it (F.F.F.); cosimo.ligori@unipr.it (C.L.); 5AST Ancona, Prevention Department, UOC Sorveglianza e Prevenzione Malattie Infettive e Cronico Degenerative, 60127 Ancona, Italy; daniel.fiacchini@sanita.marche.it; 6Post-Graduate School of Occupational Health, Università Cattolica del Sacro Cuore, 00168 Rome, Italy; nicola.magnavita@unicatt.it; 7Department of Woman, Child and Public Health, Fondazione Policlinico Universitario Agostino Gemelli IRCCS, 00168 Rome, Italy; 8Department of Interdisciplinary Medicine, Aldo Moro University of Bari, 70121 Bari, Italy; silvio.tafuri@uniba.it

**Keywords:** chickenpox, shingles, varicella zoster, varicella zoster vaccine, healthcare workers

## Abstract

Healthcare workers (HCWs) are occupationally exposed to varicella zoster virus (VZV), and their inappropriate vaccination status could contribute to an outbreak involving both professionals and the patients they care for, with a potential impact on the general population. Therefore, since 2007, the Advisory Committee on Immunization Practices (ACIP) recommends that all HCWs have evidence of immunity against varicella. The present meta-analysis was therefore designed to collect the available evidence on the seronegative status of VZV among HCWs. PubMed, Scopus, and Embase databases were searched without backward limit for articles reporting on the seroprevalence of VZV among HCWs, and all articles meeting the inclusion criteria were included in a random-effect meta-analysis model. From 1744 initial entries, a total of 58 articles were included in the quantitative analysis (publication range: 1988 to 2024), for a pooled sample of 71,720 HCWs. Moreover, the included studies reported on seroprevalence data on measles (N = 36,043 HCWs) and rubella (N = 22,086 HCWs). Eventually, the pooled seronegative status for VZV was estimated to be 5.72% (95% confidence interval [95% CI] 4.59 to 7.10) compared to 6.91% (95% CI 4.79 to 9.87) for measles and 7.21% (5.36 to 9.64) for rubella, with a greater risk among subjects younger than 30 years at the time of the survey (risk ratio [RR] 1.434, 95% CI 1.172 to 1.756). Interestingly, medical history of either VZV infection/vaccination had low diagnostic performances (sensitivity 76.00%; specificity 60.12%; PPV of 96.12% but PNV of 18.64%). In summary, the available data suggest that newly hired HCWs are increasingly affected by low immunization rates for VZV but also for measles and rubella, stressing the importance of systematically testing test newly hired workers for all components of the measles–pertussis–rubella–varicella vaccine.

## 1. Introduction

Varicella zoster virus (VZV, also known as human herpesvirus 3) is an enveloped, spherical/polygonal, double-stranded DNA virus with a genome of 124.9 kilobases (range 120 to 230 kbases; 70 open reading frames) and a diameter ranging from 180 to 200 nm [1,2,3]. VZV belongs to the order of Herpesvirales, the family Orthoherpesviridae, and it is usually included in the subfamily of alphaherpesvirus alongside herpes simplex virus 1 and 2 (HSV1 and HSV2), pseudorabiesvirus, and equine herpesvirus 1 [1,3]. Despite sharing a common ancestor, VZV has evolved several specificities, including the smallest genome of all alphaherpesvirinae [1,2].

VZV is a global and highly infectious pathogen that only infects humans, without any known animal reservoir [1,2,3,4]. Interhuman spreading usually occurs through respiratory droplets and aerosols, with subsequent primary infection in the mucosa of the upper airways [2,3,5,6]. After an incubation period ranging from 10 to 21 days (usually 14 to 16 days), VZV causes a clinical syndrome characterized by the specific skin rash also known as varicella (chickenpox) [2,5,7]. Even though skin vesicles contain a large amount of virus, being therefore considered a main source of infection [3], respiratory droplets and aerosols contain the virus 1–2 days before the onset of the typical maculopapular–vesicular rash, and 5 to 7 days thereafter, contributing its very efficient transmission, with attack rates that, in susceptible subjects, range from 61% to 100% [3,5], with a corresponding basic reproduction number (R0, i.e., the average number of secondary cases arising from the presence of one single infected case) that usually ranges from 12 to 14 [2,8]. In temperate climates and in the absence of childhood varicella vaccination, more than 90% of people are infected by VZV and develop the disease before adolescence, with the highest incidence and hospitalization rates among children aged <10 years [7,9,10]. For instance, European incidence rates range from 300 to 1291 per 100,000 people for Western Europe, to 164 to 1240 per 100,000 people for Southern Europe, and 350 per 100,000 people for Eastern Europe [7,9,10,11,12,13,14,15,16,17,18,19]. Corresponding hospitalization rates range from 1.9 to 5.8 per 100,000, peaking in the youngest age group of 0 to 12 months of age with 23 to 172 hospitalizations per 100,000 population [12,13,14,16,20,21]. Less limited evidence has been collected regarding VZV incidence rates in tropical countries [22], but available estimates suggest that the disease is acquired later in life with the highest risk of complications [9,23,24,25].

In immunocompetent and healthy subjects, primary infection from VZV usually causes a self-limited clinical syndrome [1,2,3,4,5,6,7], with long-lasting immunity, but serious complications are reported, particularly among older adults, including bacterial skin and soft tissue superinfections, occurring in 8 to 59% of all hospitalized cases, for an annual incidence rate ranging from 3.7 to 7.5 per 100,000 [13,14,15,16,21,26,27,28]; neurological complications, including aseptic meningitis, meningoencephalitis, optic neuritis, cranial nerve palsies, cerebellar ataxia, Guillain–Barré (GB) syndrome, and transverse myelitis, occurring in 4 to 61% of all hospitalized children (0.25 to 3.5 per 100,000 persons depending on the year of the study and to the geographical area) [12,13,14,21]; and respiratory complications (i.e., pneumonia and otitis media), reported in 3 to 22% of all hospitalized cases [14,16,27]. In turn, all complicated VZV infections could develop long-term sequelae (0.4 to 3.1% of all cases) [12,17,20,27,29]. The case fatality ratio (CFR) from varicella is usually estimated to be 2–4 per 100,000 cases in the general population and 0.01% to 5.4% for hospitalized cases. The most frequently reported cause of death is septicemia due to bacterial superinfections, followed by pneumonia, acute respiratory distress syndrome, myocarditis, endotoxic shock, and encephalitis [11,17,19,21,30,31,32,33]. Assuming children aged 1 to 9 years as the reference group (CFR 1/100,000), the CFR is 4 times higher for infants aged less than 1 year at the time of the infection, and 23 to 29 times higher for adults, peaking at 7% in immunocompromised individuals [19,33]. Even though most reported deaths occur in otherwise healthy individuals, several studies have pointed out that 20% to 30% of all VZV-associated deaths occur in subjects with underlying conditions, particularly immunosuppressive disorders such as acute lymphoblastic leukemia [19,33]. However, it should be noted that the CFR for varicella is reasonably underestimated, as approximately 20% of all deaths could be misclassified as HSV infection [9,10,11,31,32,34,35].

During the primary infection, even if uncomplicated, VZV can hide in the sensory nerve ganglion cells from the dorsal roots, where it remains kept in check by cell-mediated immunity. Reactivation of VZV from sensory nerve ganglia results in herpes zoster (shingles) [1,2,4], a painful, localized vesicular rash affecting the skin of the dermatome(s) containing the nerve endings of affected dorsal root ganglia. In addition to the rash, shingles can cause fever, headache, and chills and result in severe complications, including pneumonia, hearing problems, encephalitis, and most notably blindness [11,35], with disproportionally higher hospital admission rates in certain population groups (i.e., individuals of older age groups and those affected by immunosuppressive disorders) [36].

Due to the burden associated with primary infections and zoster, several therapeutic agents have been developed and eventually made available, including acyclovir for chickenpox, famciclovir, and valaciclovir for shingles, zoster-immune globulin and vidarabine for both stages [2,3,11,18,35,37]. However, the most cost-effective option remains preventive vaccination [30,38,39,40,41,42,43,44,45]. At the moment, VZV can be delivered either as a monovalent formulation or combined with vaccines for measles, parotitis, and rubella (MPR-V) [33,38,41,46]. A recombinant formulate (Recombinant Zoster Vaccine, RZV) has been recently designed in order to be delivered in adults older than 50 years of age, and a systematic review with metanalysis suggests that the RZV may be quite effective in eliciting NA against VZV [47].

Within the general population, some population groups are considered at particularly increased risk for developing primary VZV infections, if susceptible. In fact, healthcare workers (HCWs) are considered at high risk of exposure to several infectious agents due to the nature of their work, including VZV [48,49,50,51,52,53]. More precisely, because of the high transmission rate, even during the early stages of infection, HCWs could be instrumental in nosocomial transmission of VZV to susceptible persons who could develop severe complications (e.g., immunocompromised individuals, pregnant women, and infants) [48,49,50,54,55,56,57,58,59]. Therefore, vaccination of HCWs appears cost-effective and has been specifically targeted by tailored recommendations [19,33,43,44]. For example, the United States Advisory Committee on Immunization Practices (ACIP), with support from the Hospital Infection Control Practices Advisory Committee (HICPAC), since 2007 has recommended that all HCWs have evidence of immunity against varicella and that those without evidence of immunity receive two doses of varicella vaccine 4 to 8 weeks apart or, if previously received one dose, the second dose at least 4 weeks after the first dose [43,44]. Similar recommendations have been issued by several European countries. For example, the ECDC, in the guidance to varicella vaccination since 2015, stresses that vaccination of susceptible healthcare workers should be encouraged [19,33]. On the contrary, due to its specific design aimed to primarily prevent zoster rather than varicella infection, at the moment, there are no official recommendations for the delivery of the RZV in HCWs or other occupational groups [19,33,36].

Unfortunately, while obtaining high vaccination rates in HCWs should be considered a Public Health priority, several World Health Organization (WHO) regions, and most notably the European Region (EUR), are experiencing a sustained decline in vaccination rates, including those for varicella, not only in the general population but also in high-risk groups such as HCWs [38,39,42], leading some Health Authorities to the promotion of specific vaccine mandates [51,60]. In fact, available observational studies have documented quite heterogenous vaccination rates, complicating the definition of appropriate policies [19,33,61,62,63]. As a consequence, a critical appraisal of VZV vaccination rates in HCWs has a potential significance not only from an occupational health and safety point of view but also in terms of patient safety, being ultimately a considerable Public Health issue [48,61,62,63].

The present systematic review and meta-analysis have been therefore designed to achieve the following:Provide an estimate of the seroprevalence of the seronegative status for VZV in HCWs;As the VZV vaccine can be delivered either as a monovalent formulate or associated with measles, parotitis, and rubella (MPR) vaccine, estimate whether seroprevalence rates for VZV can be compared to other exanthema such as measles and rubella;As occupational physicians often are requested to perform medical surveillance without serological data, ascertain whether medical history could be a reliable hint for seroprevalence status.

## 2. Materials and Methods

### 2.1. Research Concept

The present systematic review with meta-analysis has been designed in accordance with the “Preferred Reporting Items for Systematic Reviews and Meta-Analysis” (PRISMA) statement [64] (see Appendix A), and its protocol was preventively registered in the PROSPERO database (progressive registration number CRD42024540679 https://www.crd.york.ac.uk/prospero/ (accessed on 27 May 2024)) [65].

As a preliminary step, the research concepts were defined by means of the “PECO” strategy (i.e., patient/population/problem, exposure, control/comparator, outcome) [66,67] as follows: whether HCWs (P), occupationally exposed to patient handling and aged less than 30 years (E), compared to older HCWs (C), are or not affected by a reduced occurrence of seroprevalence for varicella zoster virus (O).

### 2.2. Research Strategy

The systematic retrieval was performed across three databases (PubMed; EMBASE; and Scopus) until 30 April 2024 through the following combination of research terms (Appendix B Table A1):

#### 2.2.1. PubMed

(“Herpesvirus 3, Human” [Mesh] OR “Varicella Zoster Virus Infection” [Mesh] OR “Chickenpox” [Mesh] OR “Herpes Zoster” [Mesh]) AND (“Health Personnel” [Mesh] OR “Allied Health Personnel” [Mesh] OR “healthcare worker*” OR “health care worker*”).

#### 2.2.2. EMBASE

((“chickenpox”/exp OR chickenpox) OR “varicella zoster virus” OR “herpes zoster”) AND “health care personnel”.

#### 2.2.3. Scopus

(“Varicella Zoster Virus Infection” OR “Chickenpox” OR “Herpes Zoster”) AND (“Health Personnel” OR “Allied Health Personnel” OR “healthcare worker*” OR “health care worker*”).

No backward chronological restrictions were applied.

### 2.3. Inclusion and Exclusion Criteria

Following inclusion criteria were applied to the studies retrieved from scientific databases:(1)Reporting on HCWs directly involved in the management of patients, of any age group, in any healthcare setting (e.g., hospitals, nursing homes, etc.);(2)Providing the total number of sampled HCWs;(3)Providing the VZV seroprevalence either as crude prevalence or percent values.

Following exclusion criteria were then applied:(1)Including workers from healthcare settings not directly involved in the management of patients (e.g., laboratory workers; occupational cleaners, and hospital waste handlers, etc.);(2)Reporting on medical students from pre-clinical years;(3)Studies not including original data (i.e., systematic reviews, meta-analysis, editorial comments); case reports and/or case series; original studies still not peer-reviewed (i.e., in preprint status);(4)Not providing the total number of sampled HCWs;(5)Not providing the timeframe and/or geographical settings of the study;(6)The full text was unavailable through online repositories or by inter-library loan;(7)The main text of the relevant study was written in a language not understood by any of authors (i.e., English, Italian, German, French, Spanish, Portuguese, Turkish);(8)Not providing details on the laboratory diagnosis of VZV seroprevalence.

In case of cross-publication and/or duplicated series, only the most recent publication was included, and if it proved to be feasible, duplicated data were removed from both qualitative and quantitative analysis.

### 2.4. Selection Criteria

Articles identified through the research strategy, and consistent with inclusion and exclusion criteria, were then title- and abstract-screened to ascertain their consistency with the research question [64,68]. Studies considered relevant to the research question were then full-text screened and independently rated by two investigators (FFF and CL). Cases of disagreement were initially discussed between the investigators to obtain their consensus. When it was not reached, the chief investigator (M.R.) was involved as a third person.

### 2.5. Data Extraction

Following data were obtained from both the main text and the Appendix A (where available) and summarized:(a)Main characteristics of the study: first author’s name, year of publication, timeframe of the study; geographical settings;(b)Characteristics of the study group: sample size, baseline data of sampled HCWs (gender; age groups: proportion of individuals aged <30 y.o. vs. ≥30 y.o.), occupational groups (nurses, physicians, other professionals);(c)Outcome data: seroprevalence of VZV;(d)Appendix A: seroprevalence of rubella and measles; self-reported data on the previous infection by VZV.

### 2.6. Quality Assessment (Risk of Bias)

Risk of bias (ROB) has been defined as the likelihood that any feature of the study design or conduct may lead to misinforming results [69,70,71]. To ensure that a systematic review with/without meta-analysis is based on proper evidence, several tools have been developed for a preventive appraisal of underlying ROB, including the ROB tool from the National Toxicology Program (NTP)’s Office of Health Assessment and Translation (OHAT) (now the Health Assessment and Translation (HAT) group) [71,72]. Currently, OHAT ROB tool evaluates the internal validity of a given study by weighting 7 potential sources of bias (Appendix B Table A2) through a 4-point scale (range from: “definitely low”, “probably low”, “probably high”, to “definitely high”). Contrarily to other comparable instruments, OHAT ROB tool provides neither an overall rating for each study nor does it require that studies possibly associated with high or even very high ROB should be excluded from pooled analyses [72].

### 2.7. Data Analysis

#### 2.7.1. Descriptive Analysis

As a preliminary step, the proportion of HCWs sampled for VZV, measles, and rubella over the original population was calculated for each study. According to our main objective, seroprevalence for VZV was then calculated and reported as the percent proportion of HCWs with anti-VZV antibodies over the total of sampled HCWs. Seroprevalence rates for rubella and measles were similarly calculated. VZV seroprevalence estimates were then calculated by gender, age groups, and occupational groups where allowed by reporting strategy of the parent studies.

Risk ratios (RR) for seronegative (i.e., naïve) status with their corresponding 95% confidence intervals (95% CI) were then calculated by assuming the following groups as the reference ones: females (vs. males), age < 30 years (vs. ≥30 years), studies performed after 2020 (vs. before 2000; 2000 to 2009; 2010 to 2019), studies performed in the European WHO region (EUR; vs. Eastern Mediterranean Region [EMR]; South-Eastern Asia Region [SEAR]; Western Pacific Region [WPR]; American Region [AMR]); VZV (vs. measles, rubella).

#### 2.7.2. Diagnostic Accuracy of Medical History

The accuracy of self-reported medical history in documenting a previous VZV infection and/or the vaccination status was measured by calculating corresponding sensitivity (Se), specificity (Sp), positive and negative predicted value (PPV and PNV), diagnostic odds ratio (DOR), accuracy, and Cohen’s “kappa”. The following working definitions were applied: Se was defined as the proportion of subjects effectively immunized against VZV among people claiming any previous encounter with VZV or VZV vaccine, and Sp was defined as the proportion of subjects not effectively immunized against VZV among people denying any previous encounter with VZV or VZV vaccine. DOR expresses how much greater the odds of having the assessed condition (i.e., VZV seroprevalence) are for the people with a reported encounter with the pathogen in terms of primary infection or vaccine than for the people with a negative statement. It is a single measure of diagnostic test performance—in this case, the medical history.

Cohen’s kappa coefficient measures inter-rater reliability (and also intra-rater reliability) or agreement for qualitative (i.e., categorical) items. In other words, in the present study, it measures the agreement between medical history and laboratory specimens. Cohen’s kappa values are usually categorized as follows: kappa < 0.600 “weak” to “none” agreement; 0.600 ≤ kappa < 0.799 “moderate” agreement; 0.800 ≤ kappa < 0.900 “strong” agreement, and kappa ≥ 0.900 “almost perfect” agreement.

#### 2.7.3. Meta-Analysis

In the present meta-analysis, both pooled estimates for both seroprevalence and RR, as well as pooled Se, Sp, PPV, PNV, Cohen’s kappa, and DOR were calculated by means of a random-effect model (REM) from retrieved studies. For DOR, a correction factor of one-half was added to each cell to avoid calculation problems by having a value of zero in the 2 × 2 table. All results were reported as point estimates with their 95% CIs. In order to cope with the presumptive heterogeneity of relevant studies, in terms of sampling strategy and diagnostic option, the REM was preferred over the fixed-effects model, as it is considered more effective in dealing with the genuine differences underlying the results of the studies [73,74].

The performance of the medical history in recognizing naïve cases was eventually ascertained by plotting accuracy from each study in a summary receiver operating characteristic (sROC) with subsequent calculation of the corresponding area under curve (AUC). In the analyses, an AUC < 0.5 identified inadequate accuracy, with AUC values ranging between 0.500 to 0.749 suggesting a limited accuracy, 0.750 to 0.919 good accuracy, 0.920 to 0.959 very good accuracy, and >0.960 excellent accuracy [75].

The inconsistency of the estimates between the included studies was quantified by means of the I^2^ statistic as the percentage of total variation across studies likely due to heterogeneity (i.e., their underlying genuine differences) rather than chance [69]. I^2^ values ranging between 0 to 25% were associated with low heterogeneity; 26% to 50% with moderate heterogeneity, while I^2^ values ≥ 50% were associated with substantial heterogeneity. Because of the presumptively small size of the meta-analyses, with likely underestimation of actual heterogeneity by point estimate of I^2^, 95% CIs were provided [69].

In order to cope with potential uneven sample and effect size, sensitivity analysis (i.e., the study of how the uncertainty in the output of a mathematical model or system can be apportioned to different sources of uncertainty in its inputs) was performed to evaluate the effect of each study on the pooled estimates by excluding one study at a time.

Publication bias was assessed through the calculation of contour-enhanced funnel plots. As publication bias is suggested by asymmetry of funnel plots; after the visual appraisal, Egger’s test was performed as a confirmatory test for all outcomes with three or more included studies [64,76]. Small-study bias was similarly assessed by generating corresponding radial plots, with their visual appraisal for the uneven clustering of individual estimates across the regression line. A *p*-value < 0.05 was considered statistically significant for both publication and small-study bias.

All calculations were performed by means of R (version 4.3.1) [77] and Rstudio (version 2023.06.0 Build 421; Rstudio, PBC; Boston, MA, USA) software by means of the packages: *meta* (version 7.0), *fmsb* (version 0.7.5), and *nsROC* (version 1.1). The Prisma2020 flow diagram was designed by means of the PRISMA2020 package [78].

## 3. Results

### 3.1. Summary of Retrieved Studies

As shown in Figure 1, a total of 1744 entries were initially retrieved (i.e., 939 from EMBASE, 53.84%; 482 from Scopus, 27.64%; 323 from PubMed, 18.52%). Of them, 1121 were cross-duplicated (64.28%), while 212 records were from languages not understood by the study participants (12.16%). As a consequence, a total of 411 (23.87%) studies were screened by title and abstract, with the subsequent removal of 301 entries (17.26%). Among the 110 articles that were then sought for retrieval, 58 were ultimately retained (3.33%) and included in the present meta-analysis [46,55,57,58,61,62,63,79,80,81,82,83,84,85,86,87,88,89,90,91,92,93,94,95,96,97,98,99,100,101,102,103,104,105,106,107,108,109,110,111,112,113,114,115,116,117,118,119,120,121,122,123,124,125,126,127,128,129].

The main characteristics of the included studies are summarized in Table 1. Briefly, the studies reported on a timeframe ranging from 1988 [112] to 2024 [114], with four studies including samples collected during and after the SARS-CoV-2 pandemic [107,114,119,129]. The sample size ranged from 73 HCWs [128] to 10,576 [119]. Overall, the large majority of included studies were full papers from peer-reviewed medical journals, with a single preprint report [107], and two conference reports [79,126]. The majority of studies were from Turkey (7 out of 58, 12.07%) [81,84,88,94,106,117,128], followed by Japan [83,103,104,108,126] and Spain [95,98,100,115,121] with 5 studies each, as well as by Italy [86,97,109,114], Korea [105,110,119,129], and the USA [57,58,62,112]. Most of included studies (i.e., 47 out of 58, 81.03%) were based on ELISA [46,55,57,61,62,63,79,80,81,82,83,84,85,86,88,89,90,91,93,94,95,96,97,98,99,100,101,102,107,109,110,111,112,113,114,115,116,117,119,121,122,123,124,125,126,127,128], with 8 further reports based on EIA [87,92,103,104,106,108,118,120], 3 on CLIA [58,105,129], and a single entry on latex agglutination assay [99]. Interestingly enough, 30 studies also included data on measles [46,63,79,80,81,83,84,85,88,94,97,98,102,103,104,106,107,108,109,110,113,114,115,117,119,123,124,126,128,129] and 24 on rubella [46,63,79,80,81,83,84,85,88,97,102,103,104,106,108,109,113,114,115,117,118,123,124,126], with 23 studies reporting seroprevalence data on VZV, measles, and rubella [46,63,79,80,81,83,84,85,88,97,102,103,104,106,108,109,113,114,115,117,123,124,126].

The quality of the included studies is summarized in Figure 2, and their individual assessment is provided in Table A3. Although some studies were characterized by incomplete reporting strategy (i.e., the lack or incompleteness of demographic data), most of the included studies were reasonably characterized by high quality and a low risk of bias. In fact, only the study from Bassett et al. was affected by a definitively high risk of bias in most of the assessed domains [87].

### 3.2. Descriptive Results

Overall (Table 2), data on a total of 77,362 HCWs were collected. Of them, 38.45% were nurses, 14.50% physicians, with 32.99% other medical professionals (e.g., laboratory technicians), while in 14.06%, the job title was not provided. In a substantial share of cases, demographic data were not provided, particularly when dealing with age (41.41%) and gender (21.47%). Nonetheless, the largest share of participants was of female gender (58.70%), aged < 30 years at the enrollment (29.99%).

Detailed data on the serological tests were provided on VZV for 92.71% of participants, followed by measles (46.59%) and rubella (28.55%) (Table 3): The naïve status for VZV was identified among 9.70% of cases, compared to 10.50% for measles and 8.29% for rubella. In fact, assuming the seroprevalence rate for VZV as a reference group, HCWs exhibited an increased risk for measles seronegative status (RR 1.081, 95% CI 1.042 to 1.123) and a reduced risk for rubella (RR 0.854, 95% CI 0.813 to 0.897).

Focusing on VZV seroprevalence (Table 4), naïve status was more frequently reported in HCWs aged < 30 years compared to older ones (12.41% vs. 9.64%, RR 1.288, 95% CI 1.204 to 1.378), while no differences were found between individuals of male or female gender (9.04% vs. 9.41%, respectively; RR 1.042, 95% CI 0.966 to 1.124). Interestingly enough, studies performed after 2020 exhibited the highest rates of seronegative status (16.94%), and the risk for a seronegative status decreased with older studies (7.91% for the decade 2010–2019, followed by 6.50% for the decade 2000–2009; RR 0.467, 95% CI 0.445 to 0.490 and RR 0.384, 95% CI 0.359 to 0.410, respectively), being lowest for studies performed before 2000 (2.94%; RR 0.173, 95% CI 0.140 to 0.214).

Assuming the seronegative status from studies based in the EUR as the reference group (4.14%), a substantially increased risk for naïve status was identified among HCWs from the WPR (11.64%; RR 2.813, 95% CI 2.614 to 3.027), EMR (13.93%; RR 3.367, 95% CI 3.083 to 3.677), and SEAR (15.13%; RR 3.656, 95% CI 3.044 to 4.391), while no substantial differences were identified for studies from the AMR (3.86%, RR 0.934, 95% CI 0.755 to 1.155).

The seronegative rates for VZV and measles were significantly correlated (rho = 0.479, 95% CI 0.133 to 0.721, *p* = 0.007), while no correlation was identified for rubella (rho = 0.192, 95% CI −0.241 to 0.561, *p* = 0.369) (Appendix B Figure A1). In turn, seronegative rates for measles and rubella were not correlated to each other (r = 0.326, 95% CI −0.112 to 0.658, *p* = 0.129), and similarly, no correlation was identified between the proportion of VZV seronegative HCWs, the mean/median age of the participants (rho = −0.166, 95% CI −0.485 to 0.193, *p* = 0.349), or the sample size (rho = −0.179, 95% CI −0.096 to 0.429, *p* = 0.186). On the contrary, a positive correlation was found between the VZV seronegative rates and the proportion of individuals aged less than 30 years at the time of the study (rho = 0.492, 95% CI 0.087 to 0.757, *p* = 0.017).

### 3.3. Meta-Analysis

As summarized in Table 5, the pooled prevalence rate for VZV seronegative status was estimated at 5.72% (95% CI 4.59 to 7.10) (Appendix B Figure A2).

The majority of samples were assessed by means of ELISA (59,533 samples, 83.01%), followed by EIA (8260, 11.52%), CLIA (3033, 4.23%), and a single study (894 samples, 1.25%) on latex agglutination assay. Seronegative status was highest in studies based on EIA (7.21%, 95% CI 4.05 to 12.52), followed by ELISA (5.53%, 95% CI 4.28 to 7.13), CLIA (5.45%, 95% CI 3.34 to 8.77), and the lowest from the single study on latex agglutination assay (4.47%, 95% CI 3.22 to 7.13). Even though cumulative subgroup analysis hinted at no significant differences between groups (chi-squared 2.44, *p* = 0.489), assuming the proportion of seronegative status from ELISA-based studies as the reference category, this occurrence was less frequently reported from EIA-based studies (chi-squared 14.05, *p* < 0.001; RR 0.874, 95% CI 0.811 to 0.936) and latex agglutination assay (chi-squared 29.78, *p* < 0.001; RR 0.424, 95% CI 0.309 to 0.580), while the occurrence of seronegative status was decreased among CLIA-based studies (chi-squared 21.86, *p* < 0.001; RR 0.783, 95% CI 0.638 to 8.832). Overall, the estimates from REM were affected by substantial heterogeneity, with a point value for I^2^ equal to 98.1% and a 95% CI of 97.8% to 98.3% (tau^2^ 0.7376; H = 7.18, 95% CI 6.77; 7.61; Q 2939.67, *p* < 0.001).

Sub-analyses by age group and gender of the sampled HCWs are also reported in Table 5. In fact, the pooled prevalence of VZV seronegative status was estimated to be 9.78% (95% CI 6.91 to 13.66) in professionals aged less than 30 years at the time of the study compared to 6.31% (95% CI 4.24 to 9.28) in older ones (Appendix B Figure A3 and Figure A4). Again, estimates were affected by substantial heterogeneity (I^2^ 95.6%, 95% CI 94.2 to 96.7, and I^2^ 96.0%, 95% CI 94.8 to 97.0, respectively). The pooled seronegative status estimates were 7.39% (95% CI 5.18 to 10.44) for males and 6.98% (95% 4.95 to 9.77) for females, with substantial heterogeneity (I^2^ 91.7%, 95% CI 88.9 to 93.8, and I^2^ 97.4%, 95% CI 96.8 to 97.8, respectively) (Appendix B Figure A5 and Figure A6).

As shown in Table 6, the risk for seronegative status was eventually higher in younger than in older subjects (RR 1.434, 95% CI 1.172 to 1.755; Appendix B Figure A7), while no differences were identified between HCWs of male and female gender (RR 0.946, 95% CI 0.788 to 1.136; Appendix B Figure A8).

The pooled prevalence of seronegative status for measles was 6.91% (95% CI 4.79 to 9.87; Appendix B Figure A9) and 7.21% (95% CI 5.36 to 9.64) for rubella (Appendix B Figure A10). On the one hand, both estimates were affected by substantial heterogeneity (I^2^ 98.6%, 95% CI 98.3 to 98.8, and I^2^ 96.5%, 95% CI 95.6 to 97.2, respectively). On the other hand, the risk for seronegative status was similar for measles (RR 1.326, 95% CI 0.953 to 1.846; Appendix B Figure A9) and rubella (RR 1.335, 95% CI 0.932 to 1.910; Appendix B Figure A12) compared to VZV.

### 3.4. Diagnostic Performance of Medical History

A total of 24 studies included information about the recall of either vaccination status or previous primary infection from VZV, for a total of 3336 HCWs (i.e., 4.65% of the total sample). A pooled Se of 76.00% (95% CI 63.22 to 85.37) was then calculated with a pooled Sp of 60.12% (95% CI 48.42 to 70.76) (see Appendix B Figure A13 and Figure A14). In both cases, heterogeneity was substantial (I^2^ 99% and 97%, respectively). The corresponding PPV was estimated to be 96.12% (95% CI 92.00 to 97.16; I^2^ 99.0%, tau^2^ 3.199; Q 2207.50, *p* < 0.001) with a PNV of 18.64% (95% CI 9.73 to 32.74; I^2^ 97.5%, 95% CI 96.9 to 97.9; tau^2^ 3.381; Q 904.93, *p* < 0.001). Eventually, Cohen’s kappa was estimated to be 0.153 (95% CI 0.077 to 0.230; I^2^ 98.0%, 95% CI 97.1 to 98.9; Q 1327.499, *p* < 0.001) and with a DOR of 2.041 (95% CI 0.796 to 5.234, I^2^ 97.1%, 95% CI 96.4 to 97.6; tau^2^ 5.006; Q 781.30, *p* < 0.001) (Appendix B Figure A15).

In other words, the recalling of personal history regarding vaccination and/or previous primary infection by VZV had a doubtful discriminatory ability, with the risk of incorrectly classifying workers, ultimately classifying them and yielding more negative values among seropositive patients than among seronegative ones.

The limited reliability of recalled medical history for identifying VZV status is otherwise stressed by the ROC curve (Figure 3). As shown, an AUC of 0.642 was eventually calculated, which is a 64.2% chance that the medical professional inquiring about the status of an HCW will correctly distinguish a worker with a positive VZV status from a worker with a negative one (i.e., below the usual cut-off value of 0.7 for acknowledging a diagnostic test as acceptable).

### 3.5. Sensitivity Analysis

Sensitivity analysis was performed by removing one study at a time from the pooled estimates. Corresponding forest plots are reported in Appendix B Figure A16, Figure A17, Figure A18, Figure A19, Figure A20 and Figure A21. Briefly, when dealing with seroprevalence for VZV, measles, and rubella, the removal of a single study at a time did not affect the pooled results in terms of both pooled prevalence and heterogeneity. Similarly, sensitivity analysis of the diagnostic performance of medical history did not affect the pooled estimates.

### 3.6. Analysis of Publication Bias

Publication bias was preliminarily ascertained through the calculation of funnel plots for the prevalence of naïve status for VZV, measles, and rubella, which were then visually inspected. In each funnel plot, the sample size was plotted against the effect size: As a consequence, if the size of the sample increases, individual estimates of the effect should also converge around the true estimate(s) [63,66,73]. Funnel plots were also enhanced by adding contours of statistical significance eventually aiding in their interpretation. When dealing with a meta-analysis of prevalence rates, taking into account the underlying confidence intervals, all funnel plots appeared as substantially asymmetrical, stressing the presence of publication bias (Figure 4).

Funnel plots for the diagnostic performance on VZV serostatus were similarly calculated, being reported in Figure 5: Contrarily to the analyses on seroprevalence rates, the funnel plots on sensitivity (Figure 5a) and specificity (Figure 5b) were characterized by noticeable symmetry.

The publication bias was then assessed by Egger’s test and calculation of the corresponding radial plots (Table 7, Appendix B Figure A22).

In fact, Egger’s test suggested that all estimates but those on the sensitivity of medical history were affected by substantial publication bias.

In fact, Egger’s test suggested that all estimates but those on the sensitivity of medical history were affected by substantial publication bias, while in the corresponding radial plots, individual estimates for VZV and for the Sp of medical history were somehow clustered across the regression line, suggesting a potential small study effect. On the contrary, observations on seroprevalence for measles and rubella, as well as Sp estimates were seemingly scattered, suggesting that a small study effect could be seemingly ruled out.

## 4. Discussion

### 4.1. Summary of Main Findings

In this systematic review and meta-analysis, data from a total of 77,362 HCWs were retrieved, including VZV seroprevalence estimates for 71,720 subjects (92.71% of the original sample). Of them, 6960 (9.70%) were naïve for VZV. The risk for seronegative status was lower than that for measles (RR 1.081, 95% CI 1.042 to 1.123) but higher than that for rubella (RR 0.854, 95% CI 0.813 to 0.897). When data were pooled into the REM, the prevalence estimate for VZV seronegative status was 5.72% (95% CI 4.59 to 7.10) compared to 6.91% (95% CI 4.79 to 9.87) for measles and 7.21% (5.36 to 9.64) for rubella. The risk of seronegative status was greater among subjects younger than 30 years at the time of the survey (RR 1.434, 95% CI 1.172 to 1.756), with similar estimates in both genders (RR for female HCWs 0.946, 95% CI 0.788 to 1.136). Reported medical history of VZV infection and/or vaccination had an Se of 76.00% (95% 63.22 to 85.37), an Sp of 60.12% (95% CI 48.42 to 70.76), with a PPV of 96.12% (95% CI 92.00 to 97.16) and a PNV of 18.64% (9.73 to 32.74), with a pooled 64.2% likelihood of correctly distinguishing an HCW effectively immunized against VZV from a non-immunized HCW.

### 4.2. Generalizability and Implication for Daily Practice

A remarkable result from our metanalysis is that around 1 out of 10 HCWs aged less than 30 years at the time of the survey were likely naïve for VZV immunity. These results were, in fact, not unexpected. Since the inception of mass vaccination campaigns, the epidemiology of VZV infection has globally changed [8,38,41,131]. As recently pointed out by Huang et al. [8], the age-standardized incidence of VZV infections has progressively increased, mostly affecting the extreme of ages, including either the elderly or children under 5 years of age, that is, the age groups less likely to benefit from vaccination campaigns. In the past decades, nearly all inhabitants of Western countries had their first encounter with the pathogen under the age of 12 [19,33], developing some degree of long-lasting immunity. Conversely, since the beginning of vaccination campaigns, due to the inconsistent achievement of the target vaccination target, the circulation of the pathogen has slowed but not totally impaired [38]. For example, surveillance data from the Antelope Valley did indicate a shift in incidence peaks from 3- to 6-year-olds in 1995 to 9–11-year-olds in 2004 [38,132,133,134]. In other words, VZV seemingly walks in the footsteps of measles, with the initial successes from earlier vaccination campaigns and the subsequent occurrence of increasing vaccine hesitancy among the general population substantially paving the way for outbreaks not only involving children but also adults [135,136,137]. The increasing hesitancy towards vaccinations, including the VZV vaccine, prompted the WHO to recommend that vaccination rates above 80% should be achieved and maintained over time [8,131,132,133,134,138,139,140]. Unfortunately, not only has this target been inconsistently achieved, but even among the most successful cases, several countries still struggle to maintain the 80% target [138]. For example, VZV vaccination rates for the European Region, even where universal vaccination strategies have been implemented, range between 89.6% with one dose and 75.3% with two doses for Germany and Greece, and 60% with one dose for Turkey [39,132,138,141,142]. The suboptimal vaccination rates achieved in the general population in the last decade not only explain the higher occurrence of naïve status among individuals aged less than 30 years compared to older ones but also represent a substantial caveat for occupational medicine, as the previously unlikely hypothesis of working-age subjects being naïve for VZV has become a concrete possibility to be properly handled by Occupational Health and Safety professionals in order to guarantee both workers’ and patients’ health and safety. Where implemented by a local legal framework, Occupational Physicians (OPs), are the medical professionals responsible for health promotion in the workplace and directly contribute to immunization programs by applying and tailoring official recommendations [53,143]. For example, in most EU settings, the OP actively inquires the vaccination history of workers when biological risk is consistent with the occupational settings, recalls the vaccination status, inform the workers about the pros and cons of recommended vaccinations, and eventually either provides or promotes appropriate immunizations [53,144,145,146,147].

As most OPs from high-income countries (including both private practitioners and employees from healthcare providers) usually have no access to official vaccination registries, acquisition of the actual immunization status of recently hired HCWs can be particularly complicated, and some professionals may rely on their self-reported medical history [148,149,150]. While medical history has been proven quite effective for documenting previous vaccinations against other conditions such as tetanus [151], our data suggest that it has very limited accuracy in the ascertainment of the immunization status for VZV, at least compared to the past decades [152], and several explanations could be advocated. Firstly, since the VZV vaccine is usually delivered in childhood, adults may fail to properly recall their previous immunization [101,152], particularly in comparison with vaccines otherwise delivered in adulthood [95,151]. Second, while in the past, the diagnosis of VZV infection was simply based on the identification of the typical rash, in the post-vaccination era, clinical diagnosis has become increasingly inaccurate [153,154]. As stressed by Baum S in a recent commentary [153], nowadays, the clinical diagnosis of VZV infection could be wrong in half of cases at least, with even higher rates among fully vaccinated individuals facing breakthrough infections. This lack of familiarity has important consequences from both public and occupational health perspectives. On the one hand, it is consistent with the limited reliability of medical history, particularly when the patient/worker is asked to recall previous infections. On the other hand, by stressing the high likelihood of late or even missed diagnoses of VZV infections, it suggests that incident cases of varicella may be inaccurately handled by their healthcare providers, increasing the risk for spreading the pathogen among high-risk populations, including immunocompromised patients [153,154].

### 4.3. Implications for Practice and Policy

A rational approach to the prevention of VZV in healthcare settings depends on obtaining and maintaining high vaccination rates, particularly among individuals belonging to or working in high-risk settings (i.e., HCWs working with infants, pregnant women, immunocompromised individuals) [19,38,43,44,48]. According to our estimates, around 6% of HCWs, with an even higher proportion of younger professionals (around 10%), would be required to be vaccinated against VZV in order to be protected against the pathogen and avoid its spreading. In terms of direct costs, the conventional monovalent formulate is currently available across the EU at a cost per dose ranging from 90 EUR to 100 EUR, while the quadrivalent cost has a unitary cost ranging from 63 EUR to 100 EUR per dose. In terms of indirect costs, both options (i.e., a single antigen vaccine and combination vaccine) achieve considerable protection against VZV infection when delivered in two doses (i.e., two appointments with the responsible healthcare provider) but are considered safe and effective. On the one hand, side effects are usually limited to injection site discomfort, localized reactions, rash, and fever [140]. On the other hand, real-world effectiveness against VZV infections ranges around 90% after the conventional two-dose schedule [131]. With a vaccine failure ranging between 4% and 5% [139,155], and no evidence of waning immunity after a two-dose strategy [131], HCWs having been previously vaccinated against VZV and being able to document their status usually do not require further doses.

In this regard, as our study suggests around 1/3 of HCWs could fail to properly characterize their status, two mutually exclusive approaches can be therefore advocated for HCWs unable to provide any medical records reporting on their vaccination and/or immunization status: (a) directly delivering the VZV vaccine or (b) testing for VZV immunity, then vaccinating only individuals with proven seronegative status. Both strategies have several pros and cons that should be accurately considered. By considering a seronegative proportion of 10% among individuals aged less than 30 years, that is, most professionals at their first employment as HCWs, 9 out of 10 newly enlisted workers unable to document their status would receive two vaccination shots they do not require: Even though there is no evidence of increased risk of side effects after multiple vaccinations [8,38,41,131,139,140,155], that still means the unmotivated delivery of 18 out of 20 doses, with corresponding costs due to the vaccines and the people involved in the delivery. On the contrary, a strategy based on the preventive sampling of HCWs for their status through commercially available test kits would achieve a more rational delivery of vaccines only to naïve individuals, also reducing the consumption of healthcare resources as 9 out of 10 HCWs would reasonably require only the uptake of blood samples avoiding further accesses. Notably, point-of-care tests for VZV have been made increasingly available in recent years [156], and their progressive introduction in daily practice has the potential to improve the preventive assessment of immunization status by medical professionals and particularly OPs during the preliminary assessment of the HCWs’ fitness to work [49,156,157,158]. Nonetheless, the performances and the accuracy of most commercially available tests are far from optimal [159,160,161], spreading some reasonable concerns about their reliability for the characterization of VZV immunization status also in terms of patients’ safety. In fact, the vaccination of HCWs against VZV has two primary targets, that is, avoiding healthcare professionals from being infected during their daily practice but also reducing the risk of the spreading of VZV from infected HCWs to the patients they care for, a requirement that urges achieving a higher vaccination status among targeted individuals than among the general population. As in the present study, HCWs with a seronegative status for VZV were often characterized by inappropriate immunity against other highly communicable vaccine-preventable diseases such as measles and rubella; our results stress the critical role of the OP in guaranteeing not only the health and safety of HCWs but also that of their patients [60,162,163,164]. Even though the parent studies we recollected did not allow our analyses to appraise the share of professionals with single and multiple naïve status, it should be stressed that the VZV vaccine can be delivered as a quadrivalent formulate that also includes vaccines against measles and rubella [43,140,165]. Therefore, an immunization strategy that prioritizes vaccination of HCWs unable to document their status on the preventive assessment of their serology could be otherwise effective when a professional is required to receive multiple catch-up vaccinations [158].

### 4.4. Limitation of Evidence

Our data are affected by several shortcomings to be carefully addressed.

First, the pooled sample was heterogenous in terms of demographics, as stressed by the proportion of individuals younger than 30 years included in the parent series that were provided [46,57,58,61,80,85,86,91,93,98,100,104,107,108,110,113,114,118,119,120,127,129], ranging between 14.57% [57] and 75.65% [129]. Being individuals aged less than 30 years at the time of their recruitment and more likely to be at their first assessment by an occupational health and safety service, studies characterized by a higher proportion of younger individuals were reasonably more likely to include a higher proportion of subjects naïve to VZV. On the contrary, HCWs with greater seniority had more opportunities to have been checked for their immunization status and eventually vaccinated by their OP. Not coincidentally, our study reported a positive correlation between the proportion of younger individuals and the prevalence of seronegative HCWs. Even though our study design deliberately excluded from the pooled estimates studies only reporting on the seroprevalence data of medical students, the design of the parent studies, and particularly their sampling strategies, could have eventually biased our estimates, possibly inflating the estimated prevalence of naïve individuals.

Second, spanning the collected articles across up to 4 decades implies that the seroprevalence studies were based on various laboratory approaches. As pointed out by Breuer et al., since 2008 [161], the plethora of different tests and testing strategies developed over the years were characterized by quite heterogenous diagnostic performances. Not coincidentally, pooled estimates were not only variables in terms of the assessed timeframe and WHO region but also and most notably due to the laboratory tests employed within the study, with the highest prevalence for seronegative status from studies based on EIA (7.21%), followed by ELISA (5.53%), CLIA (5.45%), and latex agglutination assay (4.47%). There is some evidence that studies based on ELISA could underestimate the actual seroprevalence of VZV antibodies elicited from the VZV vaccine compared to the gold standard of the fluorescent-antibody-to-membrane-antigen (FAMA) test [160,161]; we cannot rule out an extensive overestimation of the actual seronegative status among sampled HCWs. Not coincidentally, among the sampled 35 studies published after 2010 [49,52,54,55,72,74,75,76,78,79,80,87,88,92,94,95,96,98,99,100,101,102,103,104,108,109,111,112,113,115,117,119,120,121,123,151], including 52,534 workers sampled for VZV (73.25% of the pooled sample), only seven series were performed with commercial tests other than ELISA [58,104,105,106,108,118,129]. Consequently, the trend towards a progressively increasing occurrence of the VZV seronegative status could partially result from an increase in false negative cases.

Third, the clinical significance of serologic tests for the appraisal of actual VZV immunity has been extensively disputed [8,160,161]. Even though ELISA is considered highly sensitive in detecting seroconversion after either infection or primary vaccination [161], breakthrough infections have been reported after vaccination even in children with IgG values exceeding the cut-off values [148,150,152]. In other words, even among sampled HCWs with documented seroprevalence of neutralizing antibodies, there may be people who appear to be protected but are not. Moreover, it should be stressed that ELISA testing for VZV encompasses various strategies [160,161] based on the antigens from the whole lysate of VZV-infected cells (WC-ELISA), purified VZV glycoproteins (gp-ELISA) or gE proteins (gE-ELISA), double antibody sandwich competitive ELISA, and double gE antigen sandwich ELISA. Their characteristics have been recently reviewed by Pan et al. [160], and, briefly, with the notable exception of Merck gp-ELISA, all of the aforementioned are either not commercially available (gE-ELISA, double antibody sandwich competitive ELISA, double gE antigen sandwich ELISA) or not sensitive enough to measure antibody response after vaccination [160,161,166,167]. Hence, the translation of our results in the clinical practice may be not so straightforward.

Fourth, we must stress the potentially limited representativity of the pooled sample. Although the studies we eventually did collect and include in this data set mostly were of high quality, the total number of included HCWs remains relatively low compared to the whole size of the healthcare workforce and to the global burden of VZV. According to available estimates, by 2020, the global workforce stock for healthcare settings encompassed a total of 65.1 million workers (i.e., 29.1 million nurses, 12.7 million medical doctors, 3.7 million pharmacists, 2.5 million dentists, with 14.9 million additional occupations) [168]: Our study not only included a total of around 70,000 HCWs, that is, 0.01% of healthcare workforce by 2020, but given the studies published across a timeframe spanning from 1988 [112] to 2024 [114], the eventual representativity could be reasonably questioned. Moreover, VZV is a very common pathogen: According to the recent estimates from Huang et al. [8], every year, it causes around 84 million incident cases, with high heterogeneity across WHO regions [8,34,38,39,40]. For comparison, clinical syndromes due to respiratory syncytial virus, usually considered highly prevalent in the general population, are estimated in 33 million cases each year [169]. In other words, although seemingly quite large, the collected sample may be quite underpowered to be considered truly representative.

All the aforementioned limits are otherwise and somehow summarized by the high heterogeneity affecting our estimates, with I^2^ exceeding 95% not only for the seronegative rate of VZV but also for measles and rubella. Due to our research strategy, having ultimately pooled together seroprevalence rates from observational studies of heterogenous designs, settings, sample sizes, and even qualities, a REM was deliberately and preventively preferred over a fixed effect model, considered more properly fitting the high amount of heterogeneity we actually identified [73,74,170]. The other side of the coin is that in a REM meta-analysis, smaller studies retain a relatively greater weight than in a fixed-effect model, leading to potential incoherencies between crude and meta-analytical estimates [73,170,171]. Not coincidentally, substantial differences in the seronegative rates of VZV and measles were identified, with crude estimates fairly exceeding the results from the REM meta-analysis: 9.70% vs. 5.72% (95% CI 4.60 to 7.10), 10.50% vs. 6.91% (95% CI 4.79 to 9.87) for VZV and measles in crude vs. pooled estimates, respectively. While these results may appear quite confusing, it should be stressed that the meta-analytical approach has been deliberately designed to better cope with potential confounders and sources of bias associated with source studies than otherwise allowed by the simple cumulative summary of individual data [170,172]. Notably, in this specific case, the simple algebraic sum of the prevalence data would have determined the substantial underestimation of the actual seronegative rates for VZV and measles among sampled HCWs, possibly leading to improper and not cost-effective health policies and recommendations. In other words, while the high heterogeneity of source studies urges for a quite cautious appraisal of our results, the substantial differences between crude and pooled estimates reasonably stress how a REM meta-analytical approach likely represents the more appropriate way for handling the highly variable landscape of observational studies on the VZV seroprevalence in HCWs.

## 5. Conclusions

This updated review of studies on VZV in HCWs offers relevant elements for the prevention of infection in this group of professionally exposed adults. The use of anamnestic data to clarify the immune status of HCWs appears to be unreliable. The widespread phenomenon of vaccine hesitancy determines a reduction in vaccine uptakes, including VZV vaccination, and this leads to increasing rates of unprotected workers among newly hired HCWs. This makes it appropriate to test newly hired workers and invite them to get vaccinated according to recommended practice.

Moreover, the age of HCWs tends to increase in all countries of the world, often at a pace greater than that of the aging of the population. Although among elderly HCWs the percentage of unprotected individuals is low, it should not be overlooked that among the elderly, there is also a share of immunosuppressed individuals for whom an epidemic could have more serious consequences. Likewise, an intra-hospital infection generated by an HCW could have very serious consequences. Not only newly hired professionals but also elderly HCWs should therefore also be tested and possibly vaccinated. The development of new tests with a greater predictive value than the current ones and their diffusion will only clarify this strategy.

## Figures and Tables

**Figure 1 vaccines-12-01021-f001:**
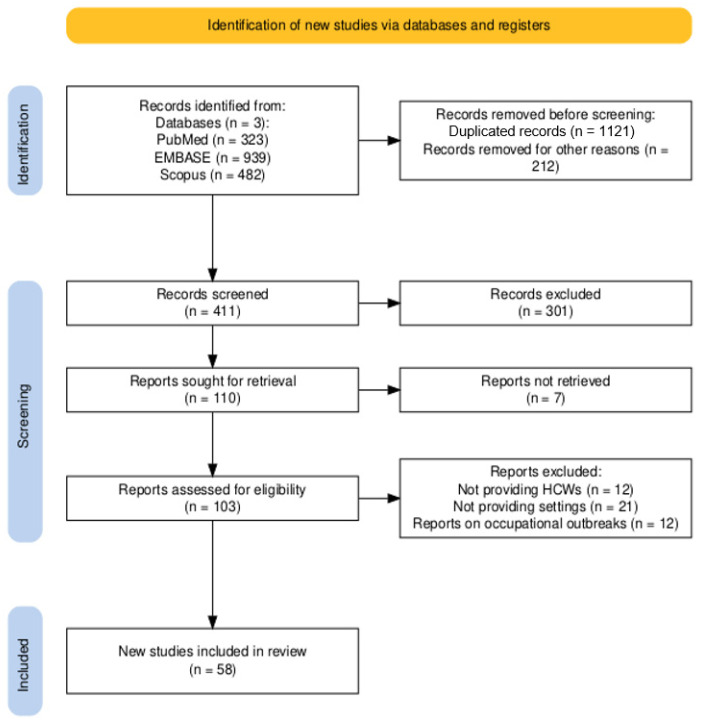
Flow chart of included studies [46,55,57,58,61,62,63,79,80,81,82,83,84,85,86,87,88,89,90,91,92,93,94,95,96,97,98,99,100,101,102,103,104,105,106,107,108,109,110,111,112,113,114,115,116,117,118,119,120,121,122,123,124,125,126,127,128,129]. Notes: HCWs = healthcare workers; other reasons = reports in languages other than English, Italian, French, German, Spanish, Portuguese, Turkish.

**Figure 2 vaccines-12-01021-f002:**
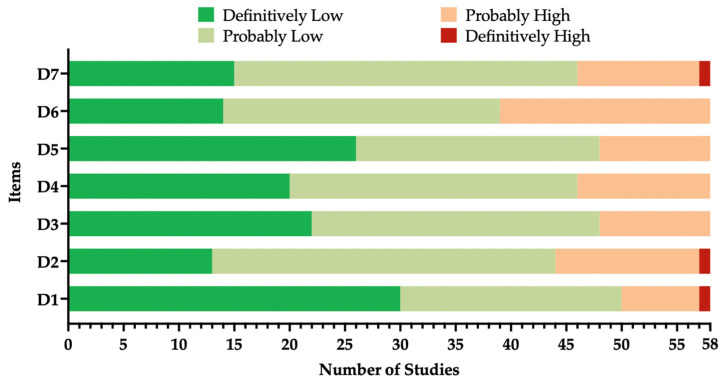
Summary of the risk of bias (ROB) estimates for observational studies [72,130]. Analyses were performed according to the National Toxicology Program (NTP)’s Office of Health Assessment and Translation (OHAT) handbook and respective risk of bias (ROB) tool.

**Figure 3 vaccines-12-01021-f003:**
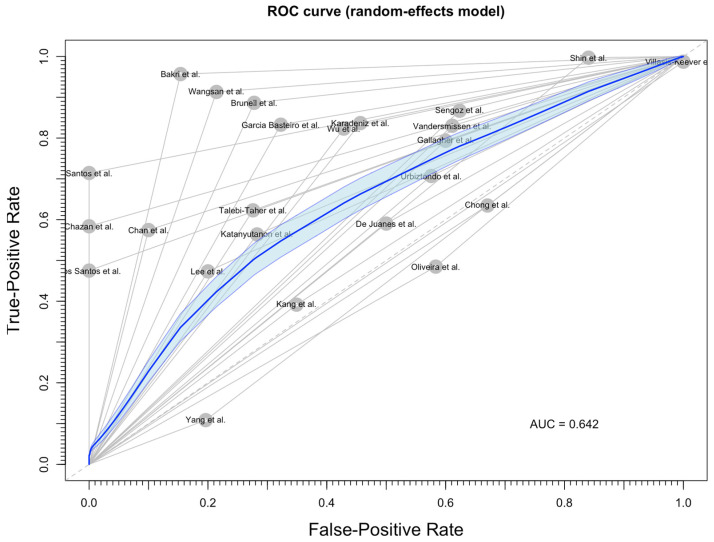
Receiver Operating Characteristics (ROCs) curve for diagnostic performance of medical history on identifying the varicella zoster virus (VZV) seropositive status in healthcare workers. A corresponding area under curve (AUC) equal to 0.642 was eventually calculated (i.e., a 64.2% chance that the medical history was able to discriminate between true positive and false positive status) [55,57,85,89,90,95,96,99,100,105,106,107,110,113,114,116,117,119,120,121,122,124,125,126].

**Figure 4 vaccines-12-01021-f004:**
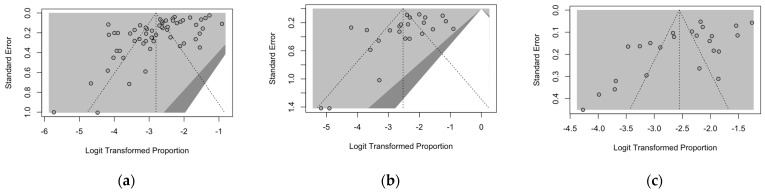
Funnel plots on the seroprevalence studies for varicella zoster virus (VZV) (**a**), measles (**b**), and rubella (**c**) in healthcare workers [46,55,58,61,62,63,79,80,81,82,83,84,85,86,87,88,89,90,91,92,93,94,95,96,97,98,99,100,101,102,103,104,105,106,107,108,109,110,111,112,113,114,115,116,117,118,119,120,121,122,123,124,125,126,127,128,129].

**Figure 5 vaccines-12-01021-f005:**
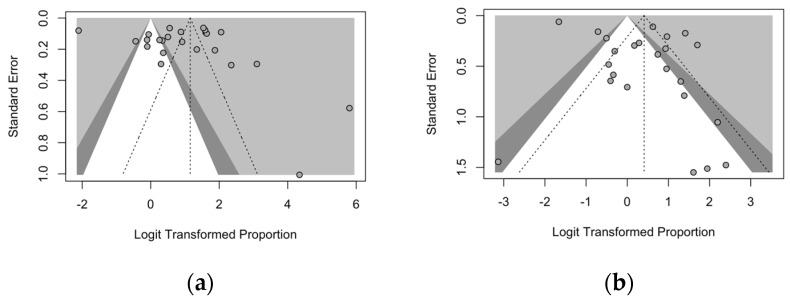
Funnel plots on the sensitivity (**a**) and specificity (**b**) of medical history for identifying healthcare workers without effective protection against varicella zoster virus (VZV; i.e., “naïve”) [55,57,85,89,90,95,96,99,100,105,106,107,110,113,114,116,117,119,120,121,122,124,125,126].

**Table 1 vaccines-12-01021-t001:** Summary of included studies. Notes: ELISA = enzyme-linked immunosorbent assay; EIA = enzyme immunoassay; CLIA = chemoluminescence immunoassay; LAA = latex agglutination assay; VZV = varicella zoster virus; N.A. = not available [46,55,57,58,61,62,63,79,80,81,82,83,84,85,86,87,88,89,90,91,92,93,94,95,96,97,98,99,100,101,102,103,104,105,106,107,108,109,110,111,112,113,114,115,116,117,118,119,120,121,122,123,124,125,126,127,128,129].

Study	Timeframe	Country	Laboratory Study	Total Sample (N)	Tested VZV (n/N, %)	Tested Measles (n/N, %)	Tested Rubella (n/N, %)	Tested Nurses (n/N, %)	Tested Physicians (n/N, %)	Age (Years)	Medical History on VZV (Yes/No)
AbdalAziz et al., 2019 [79]	N.A.	Saudi Arabia	ELISA	673	673, 100%	673, 100%	673, 100%	N.A.	N.A.	26.5 ± 5.5 (A)	No
Almuneef et al., 2006 [80]	September 2001 to March 2005	Saudi Arabia	ELISA	4006	3930, 98.10	3818, 97.80%	3972, 99.15%	913, 22.79%	267, 6.67%	N.A.	No
Alp et al., 2012 [81]	December 2010 to April 2011	Turkey	ELISA	1255	1255, 100%	1255, 100%	1255, 100%	611, 48.69%	N.A.	30 (19–60) (B)	No
Andrew et al., 2016 [46]	January 2012 to December 2013	Australia	ELISA	1901	1664, 85.53%	1779, 93.58%	1789, 94.11%	N.A.	N.A.	30 (25–38) (B)	No
Anugulruengkitt et al., 2017 [82]	May 2015 to January 2015	Thailand	ELISA	760	107, 14.08%	-	-	347, 45.66%	146, 19.21%	35.8 (28.5–47.2) (B)	No
Asari et al., 2003 [83]	April 2000	Japan	ELISA	271	271, 100%	271, 100%	271, 100%	72, 26.57%	199, 73.43%	N.A.	No
Aypak et al., 2012 [84]	N.A.	Turkey	ELISA	284	284, 100%	284, 100%	284, 100%	111, 39.08%	87, 30.63%	33.5 ± 11 (A)	No
Bakri et al., 2016 [85]	March 2011 to March 2012	Jordan	ELISA	493	493, 100%	493, 100%	493, 100%	241, 48.88%	252, 51.12%	28.8 ± 6.3 (A)	Yes
Balbi et al., 2021 [86]	January 2018 to August 2018	Italy	ELISA	840	840, 100%	-	-	297, 35.36%	463, 55.12%	36.6 (18–70) (B)	No
Bassett et al., 1993 [87]	N.A.	Hong Kong	EIA	97	70, 72.16%	-	-	97, 100%	-	N.A.	No
Behrman et al., 2013 [62]	November 2005 to May 2007	USA	ELISA	101	101, 100%	-	-	N.A.	N.A.	30 (18–70) (B)	No
Brunell et al., 1999 [57]	N.A.	USA	ELISA	1359	1359, 100%	-	-	N.A.	N.A.	N.A.	Yes
Celikbas et al., 2006 [88]	March 2005 to May 2005	Turkey	ELISA	363	363, 100%	363, 100%	363, 100%	118, 32.51%	186, 51.24%	29.1 (C)	No
Sam et al., 2008 [89]	September 2006 to April 2008	Malaysia	ELISA	88	57, 64.77%	-	-	N.A.	N.A.	26.2 (C)	Yes
Chazan et al., 2008 [90]	N.A.	Israel	ELISA	200	200, 100%	-	-	101, 50.50%	42, 21.00%	N.A.	Yes
Chodick et al., 2006 [91]	N.A.	Israel	ELISA	335	330, 98.51%	-	-	188, 56.12%	147, 43.88%	41.96 ± 12.0 (A)	No
Chong et al., 2004 [92]	September 1997 to February 1998	Singapore	EIA	2284	2284, 100%	-	-	1325, 58.01%	241, 10.55%	N.A.	No
Chong et al., 2023 [93]	August 2011 to July 2017	Taiwan	ELISA	2406	2406, 100%	-	-	959, 39.86%	366, 15.21%	N.A.	Yes
Ciliz et al., 2013 [94]	November 2011 to July 2012	Turkey	ELISA	309	309, 100%	309, 100%	-	66, 21.36%	151, 48.87%	33.8 ± 7.6 (A)	No
De Juanes et al., 2005 [95]	March 2003 to August 2003	Spain	ELISA	93	93, 100%	-	-	N.A.	N.A.	30.6 ± 4.0 (A)	Yes
Dos Santos et al., 2008 [96]	September 2002 to November 2002	Brazil	ELISA	215	215, 100%	-	-	134, 62.33%	55, 25.58%	35.3 (20.7–64.0) (B)	Yes
Fedeli et al., 2002 [97]	September 1998 to February 2002	Italy	ELISA	333	333, 100%	333, 100%	333, 100%	203, 60.96%	25, 7.51%	38 (23–60) (B)	No
Fernandez-Cano et al., 2012 [98]	January 2006 to December 2008	Spain	ELISA	2752	2511, 91.24%	2528, 91.86%	-	1014, 36.85%	632, 22.97%	42.9 ± 11.8 (A)	No
Gallagher et al., 1996 [99]	January 1990 to Dec. 1994	UK	LAA	894	894, 100%	-	-	N.A.	N.A.	N.A.	Yes
Garcia Basteiro et al., 2011 [100]	November 2000 to September 2001	Spain	ELISA	1111	1111, 100%	-	-	412, 37.08%	270, 24.30%	32.2 ± 9.2 (A)	Yes
Gorny et al., 2015 [101]	2009 to 2014	Singapore	ELISA	6701	3906, 58.29%	-	-	2221, 33.14%	124, 1.85%	N.A.	No
Guanche Garcell et al., 2016 [102]	August 2012 to December 2015	Qatar	ELISA	705	705, 100%	705, 100%	705, 100%	400, 56.74%	177, 25.11%	N.A.	No
Hatakayama et al., 2004 [103]	September 2002 to October 2002	Japan	EIA	877	854, 97.38%	860, 98.06%	867, 98.86%	426, 48.57%	212, 24.17%	34.4 ± 10.3 (A)	No
Kanamori et al., 2014 [104]	April 2012 to Mar 2013	Japan	EIA	243	243, 100%	243, 100%	243, 100%	72, 29.63%	75, 30.86%	N.A.	No
Kang et al., 2014 [105]	March 2008 to March 2010	South Korea	CLIA	550	550, 100%	-	-	393, 71.45%	103, 18.73%	27 (21–56) (B)	Yes
Karadeniz et al., 2020 [106]	September 2016 to September 2017	Turkey	EIA	1053	1053, 100%	1053, 100%	1053, 100%	481, 45.68%	395, 37.51%	22.3 ± 5.3 (A)	Yes
Katanyutanon et al., 2024 [107]	October 2022 to January 2023	Thailand	ELISA	266	266, 100%	266, 100%	-	N.A.	N.A.	38.3 ± 11.5 (A)	Yes
Kumakura et al., 2014 [108]	2005 to 2009	Japan	EIA	18,111	1811, 100%	1811, 100%	1811, 100%	622, 34.35%	662, 36.55%	34.3 ± 10.2 (A)	No
La Torre et al., 2022 [109]	February 2017 to Jan 2020	Italy	ELISA	1106	1101, 99.55%	1097, 99.19%	1105, 99.91%	462, 41.77%	336, 30.38%	54.1 ± 8.8 (A)	No
Lee et al., 2021 [110]	June 2019 to September 2019	South Korea	ELISA	300	300, 100%	300, 100%	-	203, 67.67%	34, 11.33%	33.3 ± 8.3 (A)	Yes
Lerman et al., 2004 [111]	N.A.	Israel	ELISA	335	335, 100%	-	-	-	-	N.A.	No
Lewy et al., 1988 [112]	1987	USA	ELISA	164	164, 100%	-	-	-	-	25 to 36 (D)	No
Oliveira et al., 1995 [113]	N.A.	Portugal	ELISA	409	409, 100%	409, 100%	409, 100%	-	-	40.9 ± 9.7 (A)	Yes
Perfetto et al., 2024 [114]	2017 to 2022	Italy	ELISA	517	517, 100%	517, 100%	517, 100%	-	-	26.3 ± 5.7 (A)	No
Rodriguez et al., 2014 [115]	January 2009 to June 2010	Spain	ELISA	1060	1060, 100%	1060, 100%	1060, 100%	616, 58.11%	261, 24.62%	40.2 ± 12.6 (A)	No
Santos et al., 2004 [116]	September 2002 to November 2002	Brazil	ELISA	215	215, 100%	-	-	-	-	33 (20–64) (B)	Yes
Sengoz et al., 2019 [117]	March 2014 to January 2015	Turkey	ELISA	384	384, 100%	384, 100%	384, 100%	202, 52.60%	65, 16.93%	32.4 ± 6.4 (A)	Yes
Shady I 2018 [118]	April 2015 to January 2016	Kuwait	EIA	1540	1540, 100%	-	1540, 100%	792, 51.43%	174, 11.30%	N.A.	No
Shin et al., 2023 [119]	2017 to 2022	South Korea	ELISA	10,576	9607, 90.84%	10278, 97.18%	-	5356, 51.43%	1862, 17.61%	N.A.	Yes
Talebi-Taher et al., 2010 [120]	February 2009 to March 2009	Iran	EIA	405	405, 100%	-	-	217, 53.58%	125, 30.86%	N.A.	Yes
Troiani et al., 2015 [58]	January 2014 to December 2014	USA	CLIA	413	413, 100%	-	-	N.A.	N.A.	N.A.	No
Tsou and Shao 2019 [61]	January 2008 to September 2018	Taiwan	ELISA	7314	7314, 100%	-	-	2826, 38.64%	1394, 19.06%	26.8 ± 8.0 (A)	No
Urbiztondo et al., 2014 [121]	June 2008 to December 2010	Spain	ELISA	644	644, 100%	-	-	249, 38.66%	191, 29.66%	N.A.	Yes
Vagholkar et al., 2008 [63]	September 2003 to July 2005	Australia	ELISA	1900	1320, 69.47%	1320, 69.47%	1320, 69.47%	N.A.	N.A.	N.A.	No
Vandermissen et al., 2000 [122]	February 1996 to June 1996	Belgium	ELISA	4923	4923, 100%	-	-	N.A.	N.A.	N.A.	Yes
Verma et al., 2022 [123]	July 2018 to December 2018	India	ELISA	160	160, 100%	160, 100%	160, 100%	31, 19.38%	106, 66.25%	30.6 ± 7.8 (A)	No
Villasis-Keever et al., 2001 [124]	March 1998 to May 1998	Mexico	ELISA	89	89, 100%	89, 100%	89, 100%	-	89, 100%	28 (23–41) (B)	Yes
Wangsan et al., 2019 [125]	January 2017 to September 2017	Thailand	ELISA	214	214, 100%	-	-	40, 18.69%	137, 64.02%	24 (24–27) (B)	Yes
Watanabe et al., 2013 [126]	2007 to 2012	Japan	ELISA	1385	1385, 100%	1385, 100%	1385, 100%	N.A.	N.A.	N.A.	No
Wu et al., 2012 [127]	N.A.	Taiwan	ELISA	3733	3733, 100%	-	-	1580, 42.33%	537, 14.39%	34.6 (18–68) (B)	Yes
Yang et al., 2019 [55]	January 2014 to December 2017	Mainland China	ELISA	1804	1804, 100%	-	-	1238, 68.63%	153, 8.48%	N.A.	Yes
Yavuz et al., 2005 [128]	January 2005 to March 2005	Turkey	ELISA	73	73, 100%	73, 100%	-	N.A.	N.A.	32.7 ± 5.4 (A)	No
Yun et al., 2022 [129]	October 2015 to October 2021	South Korea	CLIA	2070	2070, 100%	1827, 88.26%	-	8, 54.01%	473, 22.85%	N.A.	No

Notes: (A) mean ± standard deviation; (B) median and range (minimum–maximum; (C) mean; (D) range.

**Table 2 vaccines-12-01021-t002:** Summary characteristics of healthcare workers (HCWs) included in the systematic review.

Variable	No./Total	%
Sampled HCWs	77,362	100%
Age		
<30 years	23,202	29.99%
≥30 years	22,125	28.60%
Not reported	32,025	41.41%
Gender		
Male	15,344	19.83%
Female	45,407	58.70%
Not reported	16,611	21.47%
Job title		
Nurses	29,749	38.45%
Physicians	11,214	14.50%
Other	25,524	32.99%
Not provided	10,875	14.06%
Tested HCWs		
Varicella zoster virus	71,720	92.71%
Measles	36,043	46.59%
Rubella	22,086	28.55%

**Table 3 vaccines-12-01021-t003:** Prevalence of naïve status for varicella zoster virus (VZV) among healthcare workers (HCWs) included in the systematic review compared to data on measles and rubella.

Pathogen	N.	%	Naïve (n/N, %)	Risk Ratio	95% Confidence Interval
Varicella Zoster	71,720	92.71%	6960, 9.70%	REFERENCE	
Measles	36,043	46.59%	3784, 10.50%	1.081	1.042; 1.123
Rubella	22,086	28.55%	1830, 8.29%	0.854	0.813; 0.897

**Table 4 vaccines-12-01021-t004:** Prevalence of naïve status for varicella zoster virus (VZV) among healthcare workers (HCWs) included in the systematic review.

Variable		%	Naïve (n/N, %)	Risk Ratio	95% Confidence Interval
Gender	N./35,663				
Male	8867	24.86%	801, 9.04%	REFERENCE	
Female	26,796	75.14%	2522, 9.41%	1.042	0.966; 1.124
Age	N./27,891				
<30 years	14,451	51.81%	1793, 12.41%	1.288	1.204; 1.378
≥30 years	13,440	48.29%	1295, 9.64%	REFERENCE	
Timeframe of the study	N./71,720				
Before 2000	2896	4.04%	85, 2.94%	0.173	0.140; 0.214
2000–2009	15,885	22.15%	1032, 6.50%	0.384	0.359; 0.410
2010–2019	34,619	48.27%	2740, 7.91%	0.467	0.445; 0.490
2020 onwards	18,320	25.54%	3103, 16.94%	REFERENCE	
Settings of the study	N./71,447				
EUR	18,949	26.52%	784, 4.14%	REFERENCE	
EMR	7746	10.84%	1079, 13.93%	3.367	3.083; 3.677
SEAR	747	1.05%	113, 15.13%	3.656	3.044; 4.391
WPR	41,649	58.29%	4847, 11.64%	2.813	2.614; 3.027
AMR	2356	3.30%	91, 3.86%	0.934	0.755; 1.155

Note: EUR = World Health Organization, European Region; EMR = World Health Organization, Eastern Mediterranean Region; SEAR = World Health Organization, South-Eastern Asia Region; WPR = World Health Organization, Western Pacific Region; AMR = World Health Organization, American Region.

**Table 5 vaccines-12-01021-t005:** Summary of pooled prevalence estimates for naïve status in subgroups included in the analyses. Note: 95% CI = 95% confidence interval.

Pathogen	Grouping Variable	Pooled Prevalence (N Per 100 Samples, 95% CI)	τ^2^	(I^2^; 95% CI)	Q	*p* Value
VZV	All	5.719 (4.590; 7.104)	0.738	98.1% (97.8 to 98.3)	2939.67 (df = 57)	<0.001
	Age					
	<30 y.o.	9.775 (6.907; 13.660)	0.615	95.6% (94.2 to 96.7)	389.16 (df = 17)	<0.001
	≥30 y.o.	6.307 (4.239; 9.284)	0.760	96.0% (94.8 to 97.0)	428.04 (df = 17)	<0.001
	Gender					
	Male	7.386 (5.177; 10.435)	0.757	91.7% (88.9 to 93.8)	277.85 (df = 23)	<0.001
	Female	6.983 (4.948; 9.768)	0.776	97.4% (96.8 to 97.8)	868.98 (df = 23)	<0.001
Measles	All	6.906 (4.785; 9.871)	1.110	98.6% (98.3 to 98.8)	2001.61 (df = 29)	<0.001
Rubella	All	7.213 (5.359; 9.643)	0.585	96.5% (95.6 to 97.2)	656.83 (df = 23)	<0.001

**Table 6 vaccines-12-01021-t006:** Pooled risk ratio (RR) for the occurrence of naïve status in subgroups included in the analyses. Note: 95% CI = 95% confidence interval.

Grouping Variable	RR (95% CI)	τ^2^	(I^2^; 95% CI)	Q	*p* Value
Age < 30 vs. ≥30 y.o.	1.434 (1.172; 1.755)	0.122	79.9% (68.9 to 86.9)	84.41 (df = 17)	<0.001
Female vs. Male	0.946 (0.788; 1.136)	0.107	70.8% (55.8 to 80.7)	78.68 (df = 23)	<0.001
Measles vs. VZV	1.326 (0.953; 1.846)	0.727	98.1% (97.8 to 98.4)	1518.00 (df = 29)	<0.001
Rubella vs. VZV	1.335 (0.932; 1.910)	0.708	95.9% (94.8 to 96.7)	554.91 (df = 23)	<0.001

**Table 7 vaccines-12-01021-t007:** Summary of the results for Egger’s test performed on the main findings reported in the present meta-analysis. Note: VZV = varicella zoster virus; HCW = healthcare workers; SE = standard error; df = degrees of freedom.

Estimate	t	df	Bias (SE)	*p* Value
Proportion of naïve HCW, VZV	−5.90	56	−6.529 (1.106)	<0.001
Proportion of naïve HCW, measles	−1.76	28	−4.262 (2.426)	0.090
Proportion of naïve HCW, rubella	−3.46	22	−6.088 (1.759)	0.002
Sensitivity of medical history, VZV	0.44	22	1.981 (4.519)	0.665
Specificity of medical history, VZV	2.81	22	4.109 (1.462)	0.010

## Data Availability

Raw data are available on request to the corresponding author.

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
