# Peer review of "Immunity to Varicella Zoster Virus in Healthcare Workers: A Systematic Review and Meta-Analysis (2024)"

_vaccines, 2024, doi:10.3390/vaccines12091021_

Round 1

Reviewer 1 Report

Comments and Suggestions for Authors

In this manuscript, Ricco et. al employ meta-analysis to investigate to determine the varicella-zoster virus (VZV) seroprevalence rate in healthcare workers (HCW). The study is important because seronegative HCWs exposed to VZV infected patients may become infected and then transmit the disease to susceptible persons including immunocompromised patients. Therefore, all HCWs are recommended to provide evidence of VZV immunity or else become vaccinated.

The study reviewed 58 prior publications to determine the VZV seronegative status of 77,362 HCWs. In addition, the analysis compared the seronegative status of measles and rubella in this population.

The significant findings of the study were:

1)    The VZV seronegative rate was 9.7% among this population of HCWs.

2)    The VZV seronegative rate was significantly higher in subjects younger than age 30, while the seronegative rates were similar for male and female HCWs.

3)    The VZV seronegative rate was lower than that for measles (10.5%) and higher than that for rubella (8.29%).

4)    Patient medical history is unreliable for predicting VZV immune status due to uncertainty of patient recall and inaccuracies in clinical diagnosis.

The research protocol to retrieve data from appropriate journal articles from three databases (Pubmed, EMBASE, and Scopus) is well-designed in accordance with “Preferred Reporting Items for Systematic Reviews and Meta-analysis”.

The authors do a good job of explaining the limitations of the study including demographics and sample size of the population and possible shortcomings based on the various diagnostic procedures for determining VZV antibody titers in patient serum (ELISA, CLIA, latex agglutinination assay) compared to the gold standard FAMA test.   

There is some confusion when the data are pooled into the REM. In this case the VZV seronegative rate is 5.72% which is lower than the measles rate (6.91%) and also lower than the rubella rate (7.21%). The differences as calculated with the REM should be explained.

Author Response

Esteemed Reviewer,

Thank you for your comments, and for having stressed the following point:

There is some confusion when the data are pooled into the REM. In this case the VZV seronegative rate is 5.72% which is lower than the measles rate (6.91%) and also lower than the rubella rate (7.21%). The differences as calculated with the REM should be explained.

We totally agree that the aforementioned differences could have led to some misunderstanding, and we've updated the main text as follows:

All the aforementioned limits are otherwise and somehow summarized by the high heterogeneity affecting our estimates, with I2 exceeding 95% not only for the seronegative rate of VZV, but also for measles and rubella. Due to our research strategy, having ultimately pooled together seroprevalence rates from observational studies of heterogenous design, settings, sample size, and even quality, REM was deliberately and preventively preferred over a fixed effect model, as considered more properly fitting the high amount of heterogeneity we actually identified [73,74,170]. The other side of the coin is that in REM meta-analysis, smaller studies retain a relatively greater weight than in a fixed-effect model, leading to potential incoherencies between crude and meta-analytical estimates [73,170,171]. Not coincidentally, substantial differences in the seronegative rates of VZV and measles were identified, with crude estimates fairly exceeding the results from the REM meta-analysis: 9.70% vs. 5.72% (95%CI 4.60 to 7.10), 10.50% vs. 6.91% (95%CI 4.79 to 9.87) for VZV and measles in crude vs. pooled estimates, respectively. While these results may appear as quite confusing, it should be stressed that the meta-analytical approach has been deliberately designed to better cope with potential confounders and sources of bias associated with source studies than otherwise allowed by the simple cumulative summary of individual data [170,172]. Notably, in this specific case, the simple algebraic sum of the prevalence data would have determined the substantial underestimation of the actual seronegative rates for VZV and measles among sampled HCWs, possibly leading to improper and not cost-effective health policies and recommendations. In other words, while the high heterogeneity of source studies urges for a quite cautious appraisal of our results, the substantial differences between crude and pooled estimates reasonably stress how a REM meta-analytical approach likely represents the more appropriate way for handling the highly variable landscape of observational studies on the VZV seroprevalence in HCWs.

In summary, we're confident that the updated text could provide a better understanding of our study, that has been - therefore, substantially improved.

Again, we thank you.

MR on the behalf of all Authors.

Reviewer 2 Report

Comments and Suggestions for Authors

Dear authors, dear editors.

I was very happy to read the manuscript with the title "Immunity to Varicella Zoster Virus in healthcare workers: a  systematic review and meta-analysis (2024)."

It is a subject worth discussing.

The manuscript is a synthesis  of a systematic review and meta-analysis on the immunity to Varicella Zoster Virus (VZV) in healthcare workers (HCWs). The study focuses on the seroprevalence of VZV among HCWs, aiming to assess their risk of being seronegative for VZV and other exanthemas like measles and rubella. The review found that a significant portion of HCWs, particularly those under 30, are seronegative, highlighting the need for systematic testing and vaccination. The paper underscores the importance of HCWs' immunity as a public health issue due to their high risk of transmitting these viruses in healthcare settings.

As suggestions for improvement I analysed the key sections, providing some recommendations:

1. Abstract

  • Improvement: Ensure that the abstract clearly states the objective of the study, the methods used, the main findings (with numerical results if possible), and the implications of these findings.

2. Introduction

  • Improvement: The introduction lacks a clear statement of the gap in the existing literature that this study aims to fill. Please include a stronger rationale for the study by explicitly stating the research gap and how your study addresses it. Provide a brief overview of previous research to frame your study within the context of existing knowledge.

3. Methodology

  • Improvement: Ensure that all aspects of the methodology are thoroughly described, including the inclusion and exclusion criteria, statistical methods, and any potential biases that were addressed. Provide justifications for the chosen methods and any limitations.

4. Results

  • Improvement: Simplify the presentation of results where possible. Clearly differentiate between statistically significant and non-significant results. Ensure that all reported p-values are consistent and correctly interpreted.

5. Discussion

  • Improvement: Expand the discussion to compare your findings with those of other studies, discussing possible reasons for any differences. Highlight the study's strengths and limitations and suggest areas for future research. Discuss the practical implications of your findings for healthcare workers and public health policy. You may add more references, I suggest
  • Losa, L.et al. Immunogenicity of Recombinant Zoster Vaccine: A Systematic Review, Meta-Analysis, and Meta-Regression. Vaccines 202412, 527. https://doi.org/10.3390/vaccines12050527
  • Preda, M. et al. Advances in Alpha Herpes Viruses Vaccines for Human. Vaccines 2023, 11, 1094. https://doi.org/10.3390
  • Scampoli, P. et al. The Burden of Herpes Zoster on Hospital Admissions: A Retrospective Analysis in the Years of 2015–2021 from the Abruzzo Region, Italy. Vaccines 202412, 462. https://doi.org/10.3390/vaccines12050462

6. Conclusion

  • Improvement: Refine the conclusion to ensure it directly reflects the key findings and their relevance. Avoid introducing new information here; instead, focus on the study’s contributions to the field and any recommended actions based on the results.

7. References

  • Improvement: References should be up-to-date and relevant to the study. You may add more recent ones. 

8. Language and Style

  • Improvement: Review the manuscript for clarity and conciseness. Consider having a colleague or a professional editor review the manuscript for grammatical errors and readability.
  • 9. Figures and Tables

    • Improvement: Ensure that all figures and tables are clearly labeled and that they effectively complement the text. Consider whether additional visual aids could help to clarify complex results.

Comments on the Quality of English Language

I suggest a native English read  the manuscript.

Author Response

Esteemed reviewer,

thank you for your comments. We've accurately reviewed our text as follows:

  1. Abstract: we've double checked the results included in the abstract text.
  2. Introduction: the text was extensively revised in order to more accurately stress the knowledge gaps and the research needs.
  3. We've revised methods section in order to better define inclusion and exclusion criteria, statistical methods, and any potential biases to be addressed, stressing the justifications for the chosen methods and their potential limitations.
  4. Results section was simplified by removing not strictly necessary items that were moved to the appendix section
  5. The documents from Losa et al., Preda et al., and Scampoli et al. were included in both introduction and discussion
  6. We've refined the conclusion to stress the key findings and their relevance
  7. Several new references (alongside those from Losa et al, Preda et al, Scampoli et al.) were added in order to share new details on VZV
  8. Tables and Figures were double checked and simplified when possible

Again, we thank you for your collaborative approach and the suggestions you shared with us, whose implementation has reasonably improved the overall quality of our study.

Reviewer 3 Report

Comments and Suggestions for Authors

The manuscript integrated the  data about the status exposed to infection or vaccination of varicella zoster virus in population with more than 70 thousands healthcare workers and the seroprevalence data of it, and the seronegative rate in this population. Further,  in comprison  this seronegative rate of VZV to that of rubella or measles in association with comparison of seroprevalence of rubella or measles in the same population, author concluded that VZV immuunization should be increased in hired healthcare workers, as same as that of rubella and measles. The data analysis is strict and beneficial. However, as a research paper, firstly, this manuscript presented its description too specialized to let less understanding by general readers. Secondly, the manuscript used too many special tables and figures and longer description for a general conclusion. Therefore, it has not reached the standard for its publiscation in this "vaccines". 

Author Response

Esteemed Reviewer,

to begin with, thank you for your comments. We would like to stress your following concerns:

However, as a research paper, firstly, this manuscript presented its description too specialized to let less understanding by general readers. Secondly, the manuscript used too many special tables and figures and longer description for a general conclusion. Therefore, it has not reached the standard for its publiscation in this "vaccines".  

Our systematic review with meta-analysis was designed in accord to PRISMA 2020 guidelines, encompassing all available documents contributing to the definition of our research question. We agree that, in order to provide a wholesome and accurate answer to the our research question we have included a lot of items (i.e. tables and figures), that have possibly complicated the overall understanding of our key message.

As a consequence, we've reworked our main text as follows:

  • where possible, the text of results section was simplified;
  • tables and figures whose content was not strictly necessary for the understanding of our key messages were moved to appendix materials;
  • discussion and conclusions were refined in order to stress in a more accurate way our key results.

moreover, the whole of the main text has been extensively reworked in order to improve the overall quality of the English.

In summary, we think that our reworked paper could fulfil your concerns.

Thank you again for your suggestions.

MR

Round 2

Reviewer 3 Report

Comments and Suggestions for Authors

This manuscript has been improved and basically reached requirement of publication.